# A Critical Overview of FDA and EMA Statistical Methods to Compare In Vitro Drug Dissolution Profiles of Pharmaceutical Products

**DOI:** 10.3390/pharmaceutics13101703

**Published:** 2021-10-15

**Authors:** Jan Muselík, Alena Komersová, Kateřina Kubová, Kevin Matzick, Barbora Skalická

**Affiliations:** 1Department of Pharmaceutical Technology, Faculty of Pharmacy, Masaryk University, Palackého tř. 1, 612 00 Brno, Czech Republic; muselikj@pharm.muni.cz (J.M.); kubovak@pharm.muni.cz (K.K.); 2Department of Physical Chemistry, Faculty of Chemical Technology, University of Pardubice, Studentská 573, 532 10 Pardubice, Czech Republic; kevin.matzick@student.upce.cz (K.M.); barbora.skalicka@student.upce.cz (B.S.)

**Keywords:** drug dissolution, dissolution profile comparison, EMA and FDA strategy

## Abstract

A drug dissolution profile is one of the most critical dosage form characteristics with immediate and controlled drug release. Comparing the dissolution profiles of different pharmaceutical products plays a key role before starting the bioequivalence or stability studies. General recommendations for dissolution profile comparison are mentioned by the EMA and FDA guidelines. However, neither the EMA nor the FDA provides unambiguous instructions for comparing the dissolution curves, except for calculating the similarity factor *f*_2_. In agreement with the EMA and FDA strategy for comparing the dissolution profiles, this manuscript provides an overview of suitable statistical methods (CI derivation for *f*_2_ based on bootstrap, CI derivation for the difference between reference and test samples, Mahalanobis distance, model-dependent approach and maximum deviation method), their procedures and limitations. However, usage of statistical approaches for the above-described methods can be met with difficulties, especially when combined with the requirement of practice for robust and straightforward techniques for data evaluation. Therefore, the bootstrap to derive the CI for *f*_2_ or CI derivation for the difference between reference and test samples was selected as the method of choice.

## 1. Introduction

A dissolution study determines the in vitro release of an active pharmaceutical ingredient (API) from the tested dosage form in the prescribed dissolution medium within a specified time interval [1]. A detailed API dissolution profile resulting from dissolution testing represents one of the most essential dosage form characteristics. The dissolution test is routinely used to provide critical in vitro drug release information in pharmaceutical development and the quality control process [2,3]. It is applied in the development of brand-name drugs and generics as well. It plays a crucial role in comparing the dissolution profiles of different pharmaceutical products prior to starting the bioequivalence studies [3]. The dissolution test is an essential tool in assessing the quality and stability of a medicinal product [1,4,5]. The primary prerequisite for a qualified estimation of an API release is selecting the appropriate dissolution method with distinctive character, which can clearly detect the main differences in the API release from individual dosage forms, ideally under the conditions mimicking those for real drug absorption in the human or animal body [3]. The list of dissolution methods for individual APIs registered in the USA is published by the Food and Drug Administration (FDA) on its website [6].

Experimental dissolution data can be used for the prediction of drug release rate and mechanism from the dosage form. Various mathematical models [7,8,9,10] expressing the amount of API released as a function of time are commonly used for processing the dissolution data. Due to the non-linearity of the obtained dissolution curves (except for the zero-order kinetics), the use of non-linear regression analyses to evaluate the original measured data should be preferred. The dependency transformation of measured data into a linear form (e.g., power or logarithmic transformation) causes a change in the deviation size and the sum of squares of deviations is not minimized by the resulting function.

An important area in dissolution data analyses is the assessment of the similarity between the dissolution profiles. Several approaches have been developed for comparing the dissolution profiles. To this aim, the *f*_2_ method proposed by Moore and Flanner [11] is, due to its simplicity, the metric of choice by both the European Medicines Agency (EMA) and FDA [12,13]. The *f*_2_ similarity factor is used to assess the global similarity of the dissolution curves and it does not require any assumption regarding the data-generating process. If a substantial variation from batch to batch exists, the use of the *f*_2_ point estimate in comparing two drug dissolution profiles is not appropriate [14,15]. To circumvent this problem, the EMA and FDA guidelines allow other model-dependent or model-independent methods to be used for the dissolution profile comparison. The model-dependent methods fit the model represented by some mathematical function on data, as described above. The similarity region is derived based on a variation of estimated parameters from the fitted model for test units. The model-independent methods determine the similarity limits in terms of multivariate statistical distance. This article aims to provide a detailed overview of the FDA and EMA mentioned methods used to compare the dissolution profiles, including their procedures and limitations.

## 2. Dissolution Profile Comparison in the Context of the EMA and FDA Guidelines

The EMA and FDA guidelines [12,13] allow different approaches to be used for dissolution profiles comparison. The general recommendation is to use at least 12 units (e.g., a unit represents, for example, a tablet) of both the test *(T)* and reference *(R)* products to determine the similarity of the dissolution profiles. The dissolution profile similarity testing can be considered valid only if the dissolution profile has been satisfactorily characterized using a sufficient number of sampling time points. Where more than 85% of the drug is dissolved (which means the mean percent of drug released for both *(R)* and *(T)* samples) within 15 min, the dissolution profiles may be accepted as similar without further mathematical evaluation. If the previous requirement is not met, the similarity factor *f*_2_ is, due to its simplicity, the metric of choice by both the EMA and FDA [12,13]. When the ƒ_2_ statistic is not suitable, then the EMA guidelines [12] suggest that “*(…) the similarity may be compared using model-dependent methods or model-independent methods (…)*” and these alternative methods must be “*(…) statistically valid and satisfactorily justified*”. Except for the *f*_2_ statistic, several model-independent approaches have been developed and investigated, e.g., the Rescigno indices [16], ratio-test approaches [17], methods based on the analysis of variance (ANOVA) model [18,19,20], or calculation of the dissolution efficiency based on the area under the curve [21,22]. Unfortunately, the EMA and FDA guidelines [12,13] are not clear about the acceptability of the above-mentioned examples of model-independent methods. The EMA guidelines [12] only state which condition should be generally fulfilled if some alternative method is used. This condition is that the “*(…) similarity acceptance limits should be pre-defined and justified and not be greater than a 10% difference*” and that “*(…) the dissolution variability of the test and reference product should be also similar, however, a lower variability of the test product may be acceptable*”. On the other hand, the FDA guidelines [13] are more specific regarding similarity assessment based on model-dependent methods, i.e., comparison of profiles based on fitted mathematical function(s) to dissolution profiles. The following parts describe the basic approaches for comparing the dissolution profiles, an overview of statistical methods and recommendations in the context of the EMA and FDA guidelines (see Figure 1).

### 2.1. Similarity Factor f_2_

A simple model-independent approach uses a similarity factor *f*_2_ to compare the dissolution profiles [11]. The similarity factor *f*_2_ is a logarithmic reciprocal square root transformation of the sum of squared differences between the *(T)* and *(R)* profiles and it represents the measurement of similarity in the percentage (%) of dissolution between the “average” dissolution profiles, i.e.,
(1)$$f_{2}=50\cdot log\left\{100\cdot {\left[1+\frac{1}{n}\overset{n}{\underset{t=1}{\sum }}{\left(R_{t}-T_{t}\right)}^{2}\right]}^{-0.5}\right\}$$
where *n* is a number of time points and *R_t_* and *T_t_* are the mean percentages of the released drug from the *(R)* and (*T*) products, respectively, at the *t* time point, 1 ≤ *t* ≤ *n*. The logarithm in Equation (1) is decadic, i.e., with base 10.

The similarity factor reaches values from 0 to 100. The value of 100 indicates identical dissolution profiles. A negative value negligibly different from value 0 indicates maximal percentage dissimilarity between the profiles, i.e., 0% for the first product and 100% for the second product at each non-zero time point. The *f*_2_ value in the range between 50–100 suggests similarity of the dissolution profiles according to the EMA and FDA guidelines [13,23]. A more precise statement that the value of *f*_2_ has to be strictly greater than 50 for similarity is provided in the FDA guidelines [13]. Based on *f*_2_, the dissolution profiles are indeed considered dissimilar if the mean percentage of the released drug at each time point differs by at least 10% between the *(T)* and *(R)* products. In such a case, *f*_2_ is always less than 50.

The similarity factor *f*_2_ cannot be used on arbitrary dissolution data. According to the EMA and FDA guidelines [12,13], the following prerequisites must be satisfied:(1)the dissolution measurements are made under the same conditions for both products;(2)a minimum of three-time points (time zero excluded) is considered for both products;(3)the time points at which the dissolutions are measured are the same for both products;(4)at least 12 individual dosage units are used for both products;(5)not more than one mean percentage value is higher than 85% for any of the products;(6)the coefficient of variation (CV) of either product should be less than 20% at the first(7)(non-zero) time point and less than 10% at the following time points.

Other approaches to evaluate the similarity of the dissolution profiles have to be used if at least one prerequisite for the *f*_2_ calculation is not met, e.g., the CV values at some time points exceed the maximum permitted value. In this respect, the EMA guidelines [12] prefer the bootstrap method for the construction of the confidence interval (CI) for *f*_2_ (see Section 2.2). On the contrary, the FDA guidelines [13] prefer using some multivariate statistical distance (of which the Mahalanobis distance is the most common example), on the original dissolution data (see Section 2.4) or on the estimated parameters from regression models fitted to the original dissolution data (see Section 2.5).

It should be mentioned that the non-fulfillment of at least one prerequisite for using the *f*_2_ does not seem to be the only problem. The dissolution profile similarity can be concluded with the *f*_2_ even if the difference between the *(R)* and *(T)* samples in the mean percentage of the drug dissolved at some time point (or points) is greater than 10%. It is inconsistent with the similarity acceptance limits, which should “*(…) not be greater than a 10% difference*” as stated in the EMA guidelines [12], which refer to a 10% difference in individual time points [23]. Therefore, alternative statistical approaches to assess the similarity of dissolution profiles between the *(T)* and *(R)* products should be used in conjunction with the *f*_2_ to comply with the maximally 10% difference criterion. Such criterion can be an estimation of CI for the difference between the *(T)* and *(R)* products at each time point where all CIs should lie entirely within the similarity acceptance limits (see Section 2.3).

According to the FDA guidelines [13], it is also possible to use the difference factor *f*_1_ for the dissolution profile comparison. The difference factor is a sum of the absolute values for the differences between the *(T)* product and *(R)* products relative to the sum of the mean percentage of the released drug from the *(R)* product. This approach to assessing the dissolution profiles’ similarity is based on the same principle as when using the similarity factor. A calculated value of *f*_1_ in the range of 0–15 suggests the dissolution profiles similarity between the *(T)* and *(R)* products. The value 0 corresponds to identical dissolution profiles (including the special case when both products have identically zero mean percentage of the released drug at each time point). Interestingly, the *f*_1_ can have a theoretically unbounded upper limit if the *(T)* product is approaching the maximal dissolution (~100%) at each non-zero time point and simultaneously the *(R)* product is approaching minimal dissolution (~0%) at each non-zero time point. Details on the calculation of the difference factor are given in the literature [11,13].

The difference factor *f*_1_ and similarity factor *f*_2_ represent a simple and frequently used tool for comparing in vitro dissolution profiles. The use of this approach does not require deep knowledge in the field of statistics nor special software. It is the method of the first choice if the prerequisites mentioned above are satisfied. If at least one prerequisite for the *f*_2_ calculation is not met (e.g., CV values at some time points are higher than the maximum acceptable value), the EMA and FDA guidelines recommend using some alternative approach to evaluate the similarity of the dissolution profiles. The EMA guidelines [24] prefer the construction of CI for *f*_2_ based on the bootstrap method (Section 2.2). The FDA guidelines [13] prefer to use some multivariate statistical distance, e.g., the Mahalanobis distance (see Section 2.4.) or comparison of the dissolution profiles based on the regression parameters from the regression analysis of the dissolution data (see Section 2.5).

### 2.2. CI Derivation for f_2_ Based on Bootstrap

As stated previously, the bootstrap method for deriving CI for *f*_2_ is recommended by EMA [23] if the *f*_2_ statistic alone is not suitable. As described in Islam and Begum [14], bootstrap *“(…) is computationally-intensive approach to statistical inference (…) based on the sampling distribution of a statistic obtained by resampling from the data with replacement*”, which enables the deriving of “*(…) an exact sampling distribution of a statistic of interest*”. The statistic of interest is *f_2_* and based on knowledge of its sampling distribution, the bootstrap CI for *f*_2_ can be obtained. For dissolution data, including 12 profiles for both products (which is the minimum requirement on a number of profiles for using *f*_2_), it is preferable to use the so-called non-parametric bootstrap to construct CI for *f*_2_ [14].

Mendyk et al. [24] created statistical open-source software “PhEq_bootstrap version 1.2” to assess the dissolution profiles’ similarity, whose principle is based on the bootstrap methods under the EMA regulation.

An example of how to proceed in the bootstrap method is as follows [25]:(1)generate *n* bootstrap samples (e.g., 5000) by re-sampling from the original dissolution data;(2)calculate *f*_2_ for each of the *n* bootstrap samples, i.e., there are *n* new *f*_2_ values;(3)derive CI (lower and upper bounds) using the appropriate approach.

Regarding point (1), the test (reference) profiles in the bootstrap sample are re-sampled from the test (reference) profiles in the original data. Regarding point (3), Islam and Begum [14] concluded that the bias-corrected and -accelerated approach for calculating the percentiles leading to lower and upper bounds of bootstrap CI performs best. The dissolution profile similarity based on bootstrap CI for *f*_2_ is concluded when the lower limit of the bootstrap CI is entirely above 50 [23]. It also meets the more precise requirement found in the EMA guidelines [12], namely, that “*(…) f*_2_
*value between 50 and 100 (…)*” for similarity means that *f*_2_ has to be greater than 50.

Islam and Begum [14] mention that a two-sided 90% bootstrap CI for *f*_2_, preferably a two-sided 95% bootstrap CI, should be used.

The bootstrap method allows researchers to perform the comparison of highly variable data sets and obtains a reliable result due to limiting the impact of outliers. As mentioned above, this approach is supported by EMA in cases when the prerequisites for *f*_2_ calculation are not met and the use of *f*_2_ statistics alone is not suitable. The CI derivation for *f*_2_ based on bootstrap requires a more profound knowledge of statistics in comparison to the conventional approach (the *f*_2_ factor calculation), but the use of special software (e.g., statistical open-source software created by Mendyk [24]) can facilitate the calculation.

### 2.3. CI Derivation for the Difference between Reference and Test Samples

Based on a statement in the EMA guidelines [12] that “*(…) similarity may be compared using (…) the percentage dissolved at different time points (…)*”, another simple approach in statistical comparison of dissolution profiles is to calculate the CI for the difference between the *(T)* and *(R)* sample with respect to mean percentage of the released drug at each time point separately. Taking into account the statement in the EMA guidelines [12] that the difference should “*(…) not be greater than a 10% (…)*”, the dissolution profiles can be considered similar if the CI for the estimated difference between the *(T)* and *(R)* sample at each time point lies entirely within the interval (−10%, +10%). The lower and upper bound of the CI can be estimated according to the following expression:(2)$$R_{t}-T_{t}\pm Q_{p}\sqrt{\left(\frac{1}{n_{T,t}}+\frac{1}{n_{R,t}}\right)s_{t}^{2}}\;\;\;\;\;\;\;\;\;\;\;\;$$
where *R_t_* and *T_t_* are the mean percentage of the released drug for the *(R)* and *(T)* samples, respectively, at the *t* time point; *Q*_p_ is the *p*-quantile of the relevant probability distribution where number *p* is chosen from the interval (0,1) to obtain target confidence level for CI; *n_T_*_,*t*_ and *n_R_*_,*t*_ denote the number of measurements for the *(R)* and *(T)* samples, respectively, at *t*-the time point; *s_t_*^2^ is the pooled variance that estimates the population variance using the weighted average of variances *s_R_*_,*t*_^2^ and *s_T_*_,*t*_^2^ from *(R)* and *(T)* samples, respectively, at the *t*-the time point. Thus, the pooled estimate of variance is expressed by the equation
(3)$$s_{t}^{2}=\frac{\left(n_{R,t}-1\right)s_{R,t}^{2}+\left(n_{T,t}-1\right)s_{T,t}^{2}}{n_{R,t}+n_{T,t}-2}\;\;\;$$

In principle, Equation (2) is based on rewriting the test statistic for the two-sample *t*-test. In this case, quantile *Q*_p_ in (2) is the *p*-quantile of *t*-distribution with *n_T_*_,*t*_ + *n_R_*_,*t*_ − 2 degrees of freedom (a number of observations for the *(T)* and *(R)* products minus two at *t*-the time point) if the following assumptions are met:the observations within each sample and between samples are independent;the two samples follow normal distributions;the two samples being compared have the same variance.

Usually, a two-sided 95% CI for the difference is considered, corresponding to *p* = 0.975 for *Q*_p_.

The shortcoming of the two-sample *t*-test can be seen, for instance, in ignoring the dependence in the measured dissolutions of the same profile between different time points. The quantile resulting from the distribution of the test statistic for the multivariate two-sample Hotelling’s T^2^-test can be used to overcome this limitation. In the literature, the dissolution profile comparison using T^2^-statistics has been described in detail [26].

### 2.4. Mahalanobis Distance

A potential alternative to the similarity factor *f*_2_ can be the Mahalanobis distance (MD), the most common example of multivariate statistical distance.

In general, the Mahalanobis distance [27] is the distance between a point and distribution (not between two distinct points). The MD is equal to zero if the point is at the mean of the distribution and increases if the point moves away from the mean.

The assessment of similarity of the dissolution profiles between the *(R)* and *(T)* products using MD is based on the comparison of the vectors of the mean values of the drug released amount from the *(R)* and *(T)* products at given times. The MD for dissolution data of the *(R)* and (*T*) products is calculated according to
(4)$$\mathrm{MD}=\sqrt{{\left(T-R\right)}^{T}S^{-1}\left(T-R\right)}\;\;\;$$
where T and R is the vector of the mean percentage values of the released drug at given times for the *(T)* and *(R)* product, respectively; *(T − R**)^T^* is the transposition of the vector of the differences *(T − R)*; *S*^−1^ is the (existing) inverse matrix to the empirical covariance matrix *S.* It should be noted that, in practical calculations, the empirical covariance matrix *S* is based on the pooling covariance matrix for the *(R)* product with the covariance matrix for the *(T)* product. In addition, suppose there are *n* time points at which the dissolutions are measured. In that case, the vector of means and differences between the means has *n* components and the empirical covariance matrix is an *n* × *n* matrix. The empirical covariance matrix accounts for the dependencies in the dissolutions between different time points. The multiplication of vectors with the inverse of covariance matrix in Equation (4) gives MD as a scalar with non-negative value.

For the calculation of MD, the PC software (e.g., R statistical software, Python™) which has been implemented Equation (4) can be used. If the covariance matrix *S* is the identity matrix, then its inverse *S*^−1^ is the identity matrix as well and the MD reduces to the Euclidean distance. For the estimated value of MD, 90% or 95% CI can be calculated. The key and critical step for assessing the profile similarity based on the MD value is the determination of the equivalence margin θ (similarity threshold). For the calculation of θ, the vectors of differences *(T − R)* and *(T − R)*^T^ should be the vectors with the tens to comply with the condition in the EMA guidelines, i.e., that the difference should “*(…) not be greater than a 10% difference*” [12] between the *(T)* and *(R)* products at each time point. The inverse matrix *S*^−1^ in Equation (4) is calculated from the pooled covariance matrix *S* derived from the measured dissolution profiles. The dissolution profiles can be considered similar if the upper limit of CI for MD is lower than the value of equivalence margin θ. In other words, the similarity is concluded if CI for MD is fully within the interval [0, θ) where the lower limit of the interval uses the fact that MD defined in Equation (4) has the minimum value zero. It should be mentioned that some authors share the view that the equivalence margin should be a predefined fixed number [28,29] in comparison to the margin based on Equation (4), which is, apart from the chosen vectors of differences *(T − R)* and *(T − R)*^T^, influenced by the inverse of the empirical covariance matrix *S*. The current multivariate equivalence procedures based on MD were described in detail by Hoffelder [30]. Hoffelder [30] did not discuss only the use of Euclidean and MD in the context of the dissolution profile comparison. Moreover, he mentioned the critical moments on the MD in the current literature.

The EMA published a document concerning the adequacy of the MD to assess the comparability of drug dissolution profiles in 2018 [23]. In this document, the EMA prefers the bootstrap methodology to derive CI for *f*_2_ and does not recommend the MD for the dissolution profile comparison. The uncertainties mentioned above can affect this EMA recommendation, mainly because the equivalence limit is not given and the definition of profile similarity can vary among the dissolution datasets.

### 2.5. Model-Dependent Approach, Maximum Deviation Method

The similarity of the dissolution profiles in cases when the *f*_2_ metric is not suitable, can be also assessed using the model-dependent approach. This approach is based on fitting a regression model to the dissolution profiles (the most frequently used models are described in Table 1), the estimation of the model parameters and a comparison of the model parameters between the products for the similarity assessment. The fundamental steps in the dissolution profile regression analysis are summarized in detail in the article recently published by Muselík et al. [31]. For the dissolution profile comparison based on the model-dependent approach, the FDA guidelines [13] recommend to fit the model with no more than three parameters to each individual profile, apply some multivariate statistical distance (MSD) on the estimated parameters, derive 90% CI or 90% confidence region (CR) for MSD and, if the 90% CI or 90% CR lies entirely within the similarity region, then the similarity of the dissolution profiles is concluded. The EMA guidelines [12] are not specific regarding the similarity assessment, except for possible acceptability of MSD applied on estimated parameters from the Weibull model.

*M_t_* is the amount of drug released in the time *t*, *k*_0_ is the zero-order release rate constant, parameter *b* represents a non-zero dissolved drug amount *M**_t_* in the dissolution medium in time *t* = 0, *M_∞_* is the maximum amount of drug which can be released from a dosage form in infinite time, *k*_1_ is the first-order release rate constant, *k_w_* is the constant of Weibull model, the parameter *β* (Weibull model) characterizes the shape of the exponential curve, *k_KP_* is the constant of Korsmeyer–Peppas model, *k_H_* is the constant of Higuchi model, *M’_t_* is the drug amount in a dosage form in the time *t*, *k_HC_* is the constant of Hixson–Crowell model and *k_HP_* is the constant of Hopfenberg model.

However, based on statements in the FDA guidelines [13], it is unknown how to derive the similarity region for estimated model parameters. Moreover, there is also a question why the FDA guidelines [13] recommend using only point estimates of the model parameters for similarity assessment based on MSD. Still, the standard errors of point estimates are not considered.

Regarding the choice of similarity region for model parameters, the solution could be to fit all dissolution profiles of the *(R)* and *(T)* products to the same regression model (e.g., using OriginPro^®^ 9, GraphPad Prism^®^ 7). The same regression model clearly enables the comparison of the corresponding regression parameters between products. The regression model should be optimally fitted to all dissolution profiles of the *(T)* and *(R)* products at the same time points in order to use as much information as possible in the fitting process. The model should also be considered in such a form that the difference between the *(T)* and *(R)* products can be expressed. The similarity criterion for the difference in the estimated values of the model parameters between the products is then formulated based on the difference between the function values of the chosen model for the *(R)* and *(T)* profile. Paraphrasing the EMA guidelines [12,23], the difference in the percentage of the released drug between the *(T)* and *(R)* profiles should not be greater than 10% in the absolute value, i.e., |*T(t)* − *R(t)*| ≤ 10, for each time point *t* from a specific time range. In this manner, the acceptable difference between the estimated parameters is defined as a restriction on the difference between the function values at each time point.

Regarding the inclusion of the standard error, (1 − α), 100% CI for the difference between the estimated parameters of the *(T)* and *(R)* products mentioned above can be estimated according to the

difference between parameters” ± *t*_v,1–α_ * “standard error of the difference”(5)

where *t*_v,1–α_ is a (1 − α)-quantile of the *t*-distribution with *v* = *n_T_*_,*t*_ + *n_R_*_,*t*_ − 2 degrees of freedom (reflecting the total number of observations minus 2) and α is from the interval (0,1). The value of α is chosen in such a way to obtain the required confidence level for CI (usually 1 − α = 0.975 to obtain two-sided 95% CI for the difference in Equation (5)). The dissolution profiles of the *(R)* and *(T)* products can be considered similar if the CI for the difference between the *(R)* and *(T)* samples based on Equation (5) is fully within the range, reflecting, maximally, a 10% difference between the dissolution profiles, as described above.

An example can be given for the zero-order kinetic model without the intercept term, i.e., with *T(t)* = *k_T_* t and *R(t)* = *k**_R_ t (*where *k**_T_* and *k**_R_* are the release constants for the *(T)* and *(R)* products, respectively). The criterion for the difference between the parameters is |*k**_T_ − k_R_*| ≤ 10/*t*. This forms the similarity region [−10/*t*,+10/*t*] in which the CI for difference *k**_T_ − k_R_* must lie entirely. The lower and upper limits of the similarity region depend on time *t* from a specific time range and the time unit is the same as the time unit used for fitting the regression model.

If the intercept term (“*burst effect*”) is present, then CR (typically 90% or 95%) is simultaneously created for the intercept term and slope term. Consequently, the condition on similarity must be fulfilled for all combinations of values of both parameters from simultaneous CR. For example, considering a zero-order kinetic model with the intercept term, i.e., *T(t)* = *k*_0*T*_ + *k**_T_ t* and *R(t)* = *k*_0*R*_ + *k**_R_ t*, the condition on similarity is |(*k*_0*T*_
*− k*_0*R*_) + (*k**_T_* − *k**_R_*) *t*| ≤ 10 with time *t* chosen from a certain time range. The condition must be satisfied for all combinations of *k*_0*T*_, *k**_T_*, *k*_0*R*_ and *k**_R_* from the derived simultaneous CR. However, considering the intercept term rather leads to the situation where the same regression model is fitted separately to the *(T)* and *(R)* profiles. Such an approach of comparison parameters between two simple linear regression models and derivation of simultaneous CR was described [32].

It should be noted that Equation (5) is the most suitable for zero-order, Korsmeyer–Peppas, or the Higuchi model. In these models, the absolute difference between the parameters can be easily derived for a conclusion regarding similarity using the same model for all dissolution profiles of the *(T)* and *(R)* products simultaneously.

The expression of the similarity criterion for model parameters becomes more complicated if the estimated parameter of the used model is in the argument of the non-linear function. For example, in the case of the first-order kinetic model with *T(t)* = *M*_∞_ (1 − exp(–*k_T_ t*)) and *R(t)* = *M*_∞_ (1 − exp(–*k**_R_ t*)), an approximation approach has to be used to derive the similarity criterion for the difference *k**_T_ − k_R_*. Supposing that the maximum amount *M_∞_* of the released drug is the same for both products, the possible choice is to use the first-order Taylor polynomial leading to |*M*_∞_ (*k**_T_* − *k**_R_*) *t*| ≤ 10 and the similarity region (−10/(*M_∞_ t*), +10/(*M*_∞_
*t*)). This region depends not only on time *t* but also on the maximal release *M_∞_*. In this case, the corresponding CI for the difference *k**_T_ − k_R_* can be derived based on Equation (5) using the estimates and their standard errors obtained from the model fitted separately to all dissolution profiles of the *(T)* product and separately to all dissolution profiles of the *(R)* product. The disadvantage is that the similarity region for the difference *k**_T_ − k_R_* is not exactly expressed. Moreover, the derivation of simultaneous CR for the non-linear model is more complicated [32]. This difficulty is even higher if the parameters *k**_T_* and *k**_R_* are derived from separate (but the same) non-linear regression models.

To avoid complicated derivations for both the similarity region and CR on the “case-by-case basis,” an alternative method was proposed. Such method is called the maximum deviation (MAXDEV) method [33] and it is based on the assessment of the difference
(6)$$d_{\infty }=max_{t\in M}\left|T\left(t,\beta_{T}\right)-R\left(t,\beta_{R}\right)\right|\;$$
where *T(t, β**_T_)* and *R(t, β**_R_)* are parametric regression curves fitted to the samples of the *(T)* and *(R)* products, respectively; *β**_T_* and *β**_R_* are vectors of corresponding regression parameters; *t* denotes time from the interval *M*. Null hypothesis (*H*_0_, Equation (7)) against the alternative hypothesis (*H*_1_, Equation (8)) is formulated as
(7)$$H_{0}:\max_{t\in M}\left|T\left(t,\beta_{T}\right)-R\left(t,\beta_{R}\right)\right|\geq \Delta$$
(8)$$vs.\;H_{1}:\max_{t\in M}\left|T\left(t,\beta_{T}\right)-R\left(t,\beta_{R}\right)\right|<\Delta$$
where *Δ* is a pre-specified positive number. In the context of the statement from the EMA guidelines [12] related to tolerating, maximally, a 10% difference at each time point, the choice is *Δ* = 10. The formulation of *H*_0_ is that the dissolution profiles are not similar with respect to the chosen models for the *(T)* and *(R)* samples as the maximal absolute difference exceeds *Δ*. To better approximate the distribution of *d*_∞_ under *H*_0_, the CI for *d_∞_* is estimated based on the parametric bootstrap with a restriction on the estimated model parameters. If such CI for *d*_∞_ lies entirely in the interval [0, *Δ*), where the lower limit of the interval results from the fact that *d*_∞_ is defined as a non-negative value, the similarity between the dissolution profiles is concluded. The MAXDEV method has the advantage that the variability, time dependence between the measured dissolutions and the shapes of the dissolution profiles are simultaneously taken into account. This methodology seems to have the ability to overcome the issue when the fitted regression model is different for the *(T)* profiles and for the *(R)* profiles, although the authors demonstrated the performance of the MAXDEV method only on the same model chosen for both products. As the bootstrap is parametric, the bootstrap samples of dissolution profiles are based on bootstrapping from some parametric distribution. Specifically, the MAXDEV method bootstraps residuals from a normal distribution with zero mean and covariance structure which is estimated from the residuals resulting from the fitted models applied on the original dissolution data. Based on the bootstrap samples, the model parameters from the originally fitted models are estimated and the value of *d*_∞_ is calculated. Finally, the percentile method is used for finding suitable quantiles of the estimated distribution of *d*_∞_ to obtain the bootstrap CI of the requested confidence level for *d*_∞_. Moellenhoff et al. [33] tested the performance of the MAXDEV method on three real datasets and an artificial dataset. Based on the artificial dataset, it is shown that the type 1 error probability (T1EP) is controlled at 5% compared to the bootstrap method for construction of 90% CI for *f*_2_ (criterion for similarity: the lower limit of 90% CI has to be higher than 50) where T1EP is inflated (~55%). The statement about T1EP is made for *Δ* = 10 in *H*_0_ for *d*_∞_, as the authors are probably aware. It is compliant with the condition that the difference in dissolutions between the *(T)* and *(R)* products “*(…) should not be greater than a 10%*” [12]. The authors believe that the discrepancy in T1EP may be caused by the condition that *d_∞_* needs to have the difference “uniformly” less than 10% at each time point to conclude similarity, while *f*_2_ can assess similarity even if the difference is at least 10% at some time point (see also the brief mention in Section 2.1). The statistical power for the MAXDEV method was higher than 80% for values *d*_∞_ ≤ 8 and then declined (e.g., the power for *d*_∞_ = 9 was around 30%). The power for the bootstrap *f*_2_ was generally higher but at the cost of inflated T1EP and, as the authors mention, such inflation” *(…) is unacceptable from the regulatory point of view*”.

Using the model-dependent approach requires (i) the knowledge of the mathematical models suitable for describing the different types of dissolution profiles, (ii) the knowledge of the principles of non-linear regression analysis and (iii) suitable software. The critical point of this approach is the expression of the similarity criterion for model parameters, especially if the estimated parameter is in the argument of the non-linear function. The MAXDEV method offers the possibility to avoid complicated derivations for both the similarity region and CR. Moreover, this methodology’s advantage is it overcomes the issue when the fitted regression model is different for the *(T)* and the *(R)* profiles.

## 3. Conclusions

The release profiles evaluation very often involves the comparison of dissolution profiles (e.g., comparing generics with brand-name medicines, stability testing of pharmaceutical products). Several different methods suitable for comparing dissolution profiles are described in the literature. This article aims to provide an overview of the approaches that are supported by regulatory authorities. According to the FDA and EMA guidelines, the primary approach in comparing dissolution profiles is the similarity factor *f*_2_. In cases when the mean percentage of the released drug is higher than 85% within 15 min for both products, the similarity factor *f*_2_ does not need to be used and the profiles are automatically considered similar without any further mathematical evaluation. In other cases, either the similarity factor *f*_2_ (if all of its prerequisites are met) or some alternative statistical methods (if at least one prerequisite for using *f*_2_ is violated) are considered. In the context of comparing the dissolution profiles, some approaches described by EMA and FDA (e.g., Mahalanobis distance, model-dependent approach) are burdened with some limitations (e.g., the guidelines do not mention the similarity criteria). On the other hand, the bootstrap for the derivation of CI for *f*_2_ is an acceptable approach with clearly set criteria in statistical comparison of in vitro dissolution profiles [23]. An acceptable approach could also be the comparison of the difference between the *(T)* and *(R)* samples concerning the percentage dissolved at different time points [12] based on the calculation of CI and its comparison with pre-defined limits. Based on the discussion above, the CI derivation for *f*_2_ based on the bootstrap and CI derivation for the difference between reference and test samples could be considered methods of choice. These two methods meet the requirements of the regulatory authorities, overcome the difficulties in statistical evaluation, are robust and straightforward for routine use in practice and allow assessment of the different types of dissolution data.

## Figures and Tables

**Figure 1 pharmaceutics-13-01703-f001:**
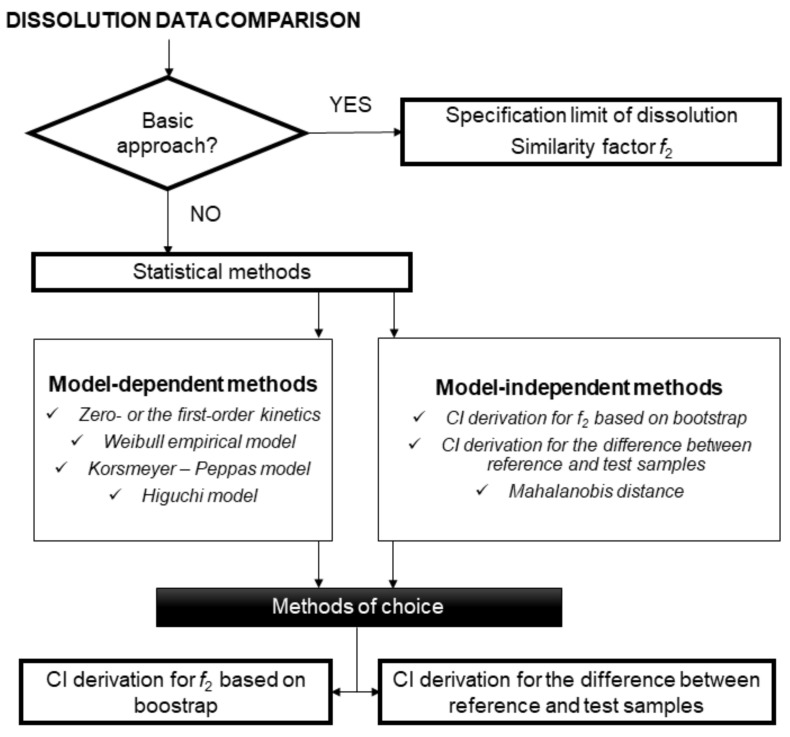
Schema of the current strategy in a dissolution data comparison by the EMA and FDA guidelines.

**Table 1 pharmaceutics-13-01703-t001:** Frequently used mathematical models to describe the drug dissolution profiles.

Model	Equation	Unit of (Rate) Constant
Zero-order	$M_{t}=k_{0}t+b$	amount of drug time^−1^ (e.g., mg h^−1^)
The first-order [32]	$M_{t}=M_{\infty }\left(1-\exp\left(-k_{1}t\right)\right)$	time^−1^ (e.g., h^−1^)
Weibull [33,34]	$M_{t}=M_{\infty }\left(1-\exp\left(-k_{w}t^{\beta }\right)\right)$	time^−β^
Korsmeyer–Peppas [35,36,37,38,39]	$\frac{M_{t}}{M_{\infty }}=k_{KP}t^{n}$	time^−n^
Higuchi [40,41,42,43,44,45,46,47]	$\frac{M_{t}}{M_{\infty }}=k_{H}\sqrt{t}$	time^−1/2^
Hixson–Crowell [48,49]	$M_{t}^{’}={\left(M_{\infty }^{\frac{1}{3}}-k_{HC}t\right)}^{3}$	drug amount^1/3^ time^−1^ (e.g., mg^1/3^ h^−1^)
Hopfenberg [50]	$\frac{M_{t}}{M_{\infty }}=1-{\left(1-k_{HP}t\right)}^{N}$	time^−1^

## Data Availability

A full list of references is compiled and attached to this manuscript.

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
