# Peer review of "A Critical Overview of FDA and EMA Statistical Methods to Compare In Vitro Drug Dissolution Profiles of Pharmaceutical Products"

_pharmaceutics, 2021, doi:10.3390/pharmaceutics13101703_

Round 1

Reviewer 1 Report

In this work, the authors reviewed the strategies used for comparison of in vitro dissolution profiles of pharmaceutical formulations. They have shown the state of art and the troubles regarding some important situations where the conventional application of similarity factor f2 analysis is not possible, leading to ambiguous instructions. The subject review is very interesting; however, some points must be clarified/revised:

1) Title should be more specific considering analysis, modeling and comparisons.

2) Lines 33-35 - The authors should consider including more other references than FDA.

3) Lines 60-390 - Considering all this information, the authors should prepare and include in the manuscript (as a figure) a flowchart/diagram showing objectively the models/equations/analysis, problem and probable ways to go, like a protocol.

4) Lines 405-410 - The authors must improve this part of conclusion.

Reviewer 2 Report

The authors make a comparison of current strategies to compare dissolution profiles of pharmaceutical products. Although the discussion is clear, I do not find in the manuscript something different from what is already published in the literature, taking into account that this literature is also easily accessible.
The description of the different strategies used to compare the solubility profiles is not enough. There should be a section where the advantages and disadvantages of each method are discussed more strictly, present some examples where they were used, present the implications of using a specific method. Mention the risks of not making an adequate comparison.
In addition, it is important to include the method section and adequately describe how the information gathering process was designed to develop this manuscript.

Reviewer 3 Report

CURRENT STRATEGY IN COMPARISON OF IN VITRO DIS SOLUTION PROFILE DATA OF PHARMACEUTICAL PROD UCTS

The paper is a simple presentation of recommended and  considered within EMA and FDA statistical methods of dissolution comparison. According to the abstract an overview of suitable statistical methods, their procedures and limitations are considered, without giving their names in the abstract. Also in the abstract is not mentioned any suggestion for  the proposed better comparison method. Therefore, minor revision is suggested.

Round 2

Reviewer 2 Report

The authors made the suggested corrections. I recommend the publication of the manuscript.

Author Response

Dear reviewer,

title of the manuscript was modified based on your suggestion.

Best regards

A. Komersová